# Colex2Lang: Language Embeddings from Semantic Typology

**Yiyi Chen**
Dep. of Computer Science
Aalborg University
Copenhagen, Denmark
yiyic@cs.aau.dk

**Russa Biswas**
FIZ Karlsruhe, AIFB
Karlsruhe Inst. of Technology
Karlsruhe, Germany
biswasrussa@gmail.com

**Johannes Bjerva**
Dep. of Computer Science
Aalborg University
Copenhagen, Denmark
jbjerva@cs.aau.dk

## Abstract

In semantic typology, colexification refers to words with multiple meanings, either related (polysemy) or unrelated (homophony). Studies of cross-linguistic colexification have yielded insights into, e.g., psychology, historical linguistics and cognitive science (Xu et al., 2020; Brochhagen and Boleda, 2022; Schapper and Koptjevskaja-Tamm, 2022; Karjus et al., 2021; François, 2022). While NLP research up until now has mainly focused on integrating syntactic typology (Naseem et al., 2012; Täckström et al., 2013; Zhang and Barzilay, 2015; Daiber et al., 2016; de Lhoneux et al., 2018; Ponti et al., 2019; Chaudhary et al., 2019; Üstün et al., 2020; Oncevay et al., 2020; Yu et al., 2021; Ansell et al., 2021; Zhao et al., 2021; Oncevay et al., 2022), we here investigate the potential of incorporating semantic typology, of which colexification is an example. We propose a framework for constructing a large-scale synset graph and learning language representations with node embedding algorithms. We demonstrate that cross-lingual colexification patterns provide a distinct signal for modelling language similarity and predicting typological features. Our representations achieve a 9.97% performance gain in predicting lexico-semantic typological features and expectantly contain a weaker syntactic signal. This study is the first attempt to learn language representations and model language similarities using semantic typology at a large scale, setting a new direction for multilingual NLP, especially for low-resource languages.[1]

## 1 Introduction

Semantic typology studies cross-lingual semantic categorization (Evans et al., 2010). The term "colexification", which encompasses both polysemy and homophony, was introduced to the field of semantic typology by François (2008). This study focuses on cross-lingual colexification patterns, where the same lexical form is used in distinct languages to express multiple concepts. For instance, *bla* in Monpa Changprong and *afu* in Rikou both express the concepts DUST and ASH (Rzymski et al., 2020).

Colexification was first used in linguistic typology to create semantic maps. Haspelmath (2003) created a semantic map with 12 languages, and François (2008) pointed out that the number of different senses increases with the number and variety of languages used. In recent years, big data, and improved data creation and curation techniques have led to the development of datasets like Concepticon (Forkel et al., 2020), and BabelNet (Navigli and Ponzetto, 2012), which make large-scale cross-lingual semantic comparisons possible. The Cross-Linguistic Colexifications (CLICS) database was created based on the Concepticon collection and is being continuously maintained. The current version[2] CLICS[3] includes 4,228 colexification patterns across 3,156 languages. In this paper, we create a synset graph based on multilingual WordNet (Miller, 1995) data from BabelNet 5.0, compare it with the concept graph extracted from CLICS[3], and explore the impact of data scope on language representation learning.

We hypothesize that language representations learned using semantic typology encapsulate a distinct language signal, and the data size of colexifications has an impact on the learned language representations and the modelled language similari-

---

[1]GitHub: https://shorturl.at/bioUZ.

[2]https://clics.clld.org/

ties. Importantly, we expect that this type of signal can be used to improve semantically oriented downstream tasks in NLP. To test this hypothesis, we propose a framework `Colex2Lang` (cf. Section 3) to learn language representations leveraging semantic typology, conduct typological feature prediction, and model language similarities. Our experiments on typological feature prediction focus on the domain of *semantic* features, so as to investigate the extent to which a semantic signal is encapsulated by our language representations.

Specifically, we make the following contributions: (i) We generate and evaluate 24 sets of language embeddings based on large-scale colexification databases, using four advanced node embeddings algorithms, i.e., Node2Vec (Grover and Leskovec, 2016), ProNE (Zhang et al., 2019), GGVec [3], and GloVe [4] (Pennington et al., 2014); (ii) we conduct thorough experiments on typological feature prediction to compare colexification-informed and more general language embeddings (Malaviya et al., 2017; Östling and Tiedemann, 2017), which provides a strong benchmark for further research; (iii) we demonstrate the usability of modelling language similarities based on colexification patterns, and argue for the potential of utilising semantic typology in NLP applications.

## 2 Related Work

**Colexification** Cross-linguistic colexifications were first formalized by François (2008) for the creation of semantic maps. Semantic maps represent the relation between recurring meaning expressions in a language graphically (Haspelmath, 2003). The basic idea underpinning this method is that language-specific patterns of colexifications indicate semantic closeness or relatedness between the meanings that are colexified (Hartmann et al., 2014). When investigated cross-lingually, colexification patterns can provide insights in various fields, such as recognizing cognitive principles (Berlin and Kay, 1991; Schapper et al., 2016; Jackson et al., 2019; Gibson et al., 2019; Xu et al., 2020; Brochhagen and Boleda, 2022), diachronic semantic shifts in individual languages (Witkowski and Brown, 1985; Urban, 2011; Karjus et al., 2021; François, 2022), and the evolution of language contact (Heine and Kuteva,

2003; Koptjevskaja-Tamm and Liljegren, 2017; Schapper and Koptjevskaja-Tamm, 2022).

Jackson et al. (2019) investigated cross-lingual colexifications in the domain of emotions and found that languages have different associations between emotional concepts. For example, Persian speakers associate the concept of GRIEF with REGRET closely whereas Dargwa speakers associate it with ANXIETY. The cultural variation and universal structure shown in the emotion semantics provide interesting insights into NLP. Di Natale et al. (2021) used colexification patterns to test whether the words linked by colexification patterns capture similar affective meanings, and subsequently expanded affective norms lexica to cover exhaustive word lists when additional data are available. Inspired by Jackson et al. (2019), Sun et al. (2021) proposed emotion semantic distance, measuring how similarly emotions are lexicalized across languages, to improve cross-lingual transfer learning performance on sentiment analysis. Bao et al. (2021) show that there exists no universal colexification pattern by analyzing colexifications from BabelNet, Open Multilingual WordNet (Bond and Foster, 2013), and CLICS[3].

Closely related to our work, Harvill et al. (2022) constructed a synset graph from BabelNet to improve performance on the task of lexical semantic similarity. Instead of modelling only word similarity using colexification patterns, we strive to model language similarity in this study and show that the language embeddings learned on colexification patterns capture a unique semantic signal compared to language embeddings encapsulating syntactical signals. Moreover, we experiment with different node embedding algorithms and compare three colexification datasets. The framework provides a strong benchmark for further investigating how semantic typological aspects of language embeddings can be leveraged in broader applications, especially for low-resource multilingual NLP.

**Node Embeddings** Node embeddings can be broadly classified into three different categories namely **(i)** matrix factorization-based models, **(ii)** random walk-based models, and **(iii)** deep neural network-based models, as discussed in (Cui et al., 2018).

In matrix factorization-based models, an adjacency matrix is used to denote the topology of a network. Matrix factorization techniques, such as Singular Value Decomposition (SVD) and Non-

---

[3] `https://github.com/VHRanger/nodevectors`
[4] `https://shorturl.at/myzKR`

negative Matrix Factorization (NMF), can be applied to address this problem. GraRep (Cao et al., 2015) considers k-hop neighbourhoods utilizing SVD of the adjacency matrix. This model often only captures small-order proximity and has a significant computational complexity for large graphs. The asymmetric transitivity is preserved by the HOPE (Ou et al., 2016) model as it converts the problem to a generalised SVD problem reducing the complexity. ProNE (Zhang et al., 2019) introduces a sparse matrix factorization to achieve initial node representations efficiently.

Random walks are used to maintain local neighbourhoods of nodes and their attributes (Newman, 2005), by increasing the likelihood of a node's neighbourhood given its embedding using the Skip-gram model (Mikolov et al., 2013). The objective behind these models is to optimize via stochastic gradient descent on a single-layer neural network, resulting in decreased computing complexity.

DeepWalk (Perozzi et al., 2014) randomly chooses a node and proceeds to walk to each neighbouring node until it reaches its maximum length (or some random length). LINE (Tang et al., 2015) aims to embed nearby vertices that either have linkages between them (optimizing for first-order proximity) or have a shared 1-hop neighbourhood (optimizing for second-order proximity). Node2vec (Grover and Leskovec, 2016) proposes a second-order random walk approach to sample the neighbourhood nodes with biasing parameters of Breadth First Search (BFS) and Depth First Search (DFS). A meta-strategy for graph embedding under recurrent construction of nodes and edges into condensed graphs with the same global structure is proposed by HARP (Chen et al., 2018). These graphs serve as source initializations for the detailed graphs that are embedded, producing appropriate node and edge embeddings as a consequence. Metapath2vec (Dong et al., 2017) is an extension of DeepWalk that formalizes meta-path-based random walks to build a node's neighbourhood, then uses a heterogeneous skip-gram model.

GGVec algorithm directly minimizes distances between the related nodes and is designed for large networks. It uses negative sampling followed by minimization loss to learn the node embeddings based on the minimal dot product of edge weights. Another node embedding model follows the word embedding model GloVe (Pennington et al., 2014) which is based on word co-occurrences and is beneficial for sparse matrices. The graph is represented by an adjacency matrix and the co-occurrence matrix is calculated using the frequency of node co-occurrences in the graph instead of word co-occurrences.

**Typological Feature Prediction** Linguistic typologists analyse languages in terms of their structural properties (Croft, 2002). As documenting and categorising such cross-lingual variation across the languages in the world is one of the core activities in typology, one of the outcomes of research in linguistic typology is large typological databases (e.g. the World Atlas of Language Structures (WALS, Dryer and Haspelmath (2013)). While such variation can be found across the spectrum of languages, the earliest work in the field largely focused on morphosyntactic properties (e.g. Greenberg (1957)), concretely looking at minimally meaning-bearing elements (morphemes), combinations thereof, and patterns of their use. For instance, well-documented features include word ordering (e.g. English is SVO, Japanese is SOV) and affixation (German uses case suffixes, Berber uses case prefixes).

Prediction of such features has gained interest in recent years (Malaviya et al., 2017; Bjerva et al., 2019a, 2020; Bjerva and Augenstein, 2021), and it has been shown that embeddings trained solely from tasks such as machine translation (Malaviya et al., 2017) or language modelling (Östling and Tiedemann, 2017) can encapsulate such features. Further analysis has shown that the nature of the underlying data used to generate language embeddings can have a significant impact on what features are encapsulated (Bjerva and Augenstein, 2018a,b; Bjerva et al., 2019b), and even that such representations contain typological generalisations (Östling and Kurfalı, 2023). Previous work is limited in that it almost exclusively relates to *morphosyntactic* typological features. In this work, we aim to present initial evidence that a lexico-semantic signal can be better learned from a lexico-semantic data source.

## 3 Colex2Lang

To better understand and leverage semantic typological features in NLP, we propose a framework – `Colex2Lang` (Fig.1) – to model language representations based on a synset graph,

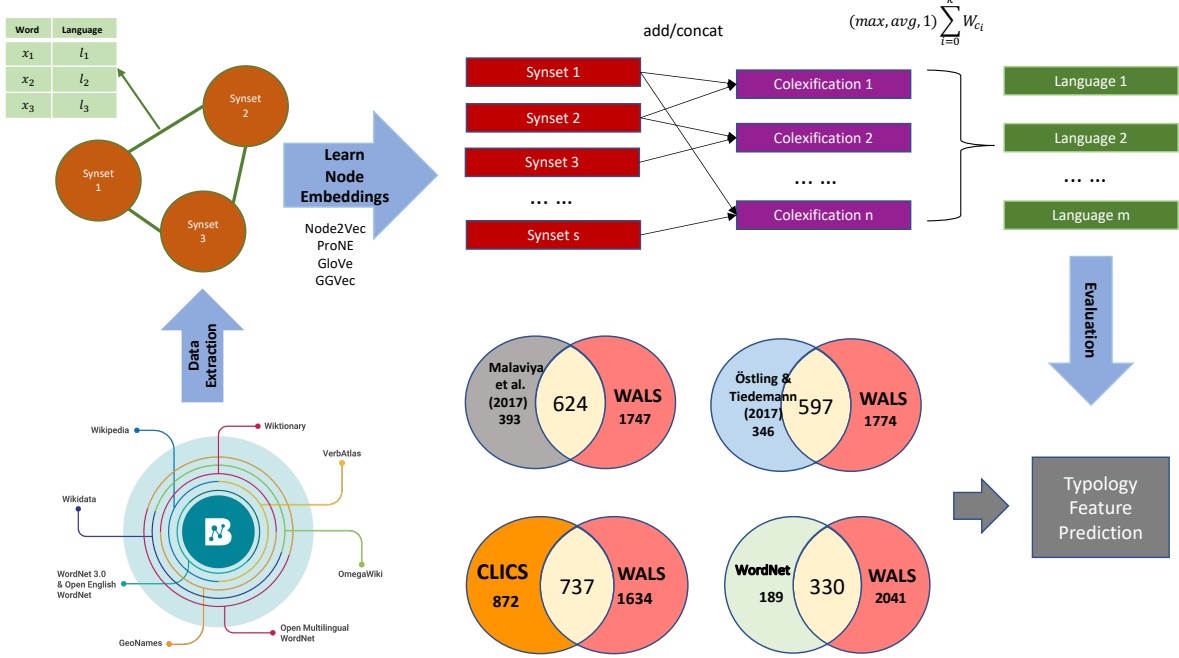

Figure 1: Framework for Colex2Lang. The numbers in the Venn diagrams denote the number of languages.

created from large-scale databases, and evaluate and analyse the language representations. The framework `Colex2Lang` is composed of the following steps:

**Building the Synset/Concept Graph**

We use WordNet synsets, extracted from BabelNet 5.0, to create a synset graph. The construction of a synset graph is formalized in Harvill et al. (2022) (see details in Appendix A). In BabelNet, every synset is either a concept or a named entity or has no type. The dataset with only concepts and with all types of synsets from WordNet are created, denoted as "WordNet Concept" and "WordNet" respectively. In analogy, CLICS[3] provides a graph of concepts, from which we extracted the colexification patterns for all the languages having an ISO 639-2 code [5], denoted as CLICS. In this study, we use "concept" and "synset" interchangeably. The statistics of the curated datasets are shown in Table 1.

**Creating Synset and Language Embeddings**
To capture the semantic associations among synsets, given the synset/concept graph $G_s$, we train synset embeddings using four node embedding algorithms and compare them: Node2Vec, ProNE, GGVec and GloVe. Given the learned

synset embeddings, we obtain the colexification embeddings $W_c$ by concatenating or summing the synset embeddings $W_s$; thereafter, the language embeddings $W_l$ are created by summing, averaging or max-pooling the consisting colexification embeddings $W_c$. For example, if the synset embeddings are trained with ProNE, and are concatenated to compose colexification embeddings, which in turn are max-pooled to obtain language embeddings, we denote the language embeddings as $W_{prone\_concat+max}$.

**Evaluation**  To obtain insights into the learned language embeddings based on the colexification patterns, such as which aspect of language these language embeddings capture and to what extent they can assist in improving NLP tasks, we conduct typological feature prediction and analyse the results in depth. Furthermore, the language embeddings are used to model language similarities, to demonstrate the potential of applications in contributing to cross-lingual transfer learning.

## 4   Experiments

**Datasets**  To better understand the impact of data scope on the NLP task performance, we curate three different datasets, i.e., WordNet, WordNet Concept, and CLICS, as described in Sec-

[5]https://shorturl.at/hBCF0

| Dataset | #$(C, X, L)$ | Colexifications ($C$) | Lexicalizations ($X$) | Synsets / Concepts | #Language ($L$) (Pair) |
|---|---|---|---|---|---|
| WordNet | 6,199,897 | 2,525,591 | 974,346 | 105,827 | 519 (134421) |
| WordNet Concept | 6,075,413 | 2,486,485 | 920,031 | 99,817 | 519 (134421) |
| CLICS | 68,560 | 4,228 | 53,259 | 1,647 | 1609 (332783) |

Table 1: Statistics of Colexification Datasets

| ∩WALS | #Language | Lexicon | | | Complex Sentences | | | Nominal Categories | | | Simple Clauses | | | 10 Feature Areas | | |
|---|---|---|---|---|---|---|---|---|---|---|---|---|---|---|---|---|
| | | #F | #V | #D | #F | #V | #D | #F | #L | #D | #F | #V | #D | #F | #V | #D |
| CLICS | 737 | 13 | 4 | 93 | 7 | 4 | 86 | 29 | 5 | 145 | 26 | 4 | 142 | 188 | 9 | 288 |
| WordNet (Concept) | 330 | 13 | 2 | 58 | 7 | 4 | 56 | 29 | 5 | 92 | 26 | 4 | 89 | 185 | 8 | 166 |
| Malaviya et al. (2017) | 624 | 13 | 4 | 92 | 7 | 4 | 63 | 29 | 5 | 112 | 26 | 4 | 117 | 190 | 9 | 238 |
| Östling and Tiedemann (2017) | 597 | 13 | 4 | 85 | 7 | 4 | 60 | 29 | 5 | 103 | 26 | 4 | 109 | 190 | 9 | 219 |

Table 2: Statistics of Typology Feature Prediction Datasets. Under each feature area and in all ten feature areas, #F represents the total number of features, #V represents the average number of feature values, #D represents the average number of data samples.

tion 3. As shown in Table 1, there are far more (unique) colexification patterns in fewer languages in WordNet-based datasets compared to CLICS, i.e., 6 Mio colexifications with more than 2 Mio unique colexification patterns constructed from 105K synsets in 330 languages, and 68K colexifications with 4K unique colexification patterns from 1,647 concepts across 1609 languages, respectively. The synset embeddings are trained separately on the three datasets with four different node embeddings algorithms, and the language embeddings are composed accordingly, as described in Section 3. Eventually, for each dataset, there are 24 sets of colexification-informed language embeddings [6].

We hypothesize that (i) the colexification-informed language embeddings capture a unique language aspect and (ii) the language embeddings learned on large-scale WordNet datasets present stronger semantic typological signals than the ones trained on CLICS. To test this, we rely on WALS v2020.3 [7], the most used and comprehensive database for typology feature prediction, consisting of 2,662 languages. For our experiment, we extract language data from WALS by ISO 639-2 codes, resulting in a dataset of a total of 2,371 languages, 192 typological features across ten feature areas, i.e., phonology, morphology, lexicon, complex sentences, nominal categories, nominal syntax, simple clauses, verbal categories, word order, and other. To test hypothesis (i), the language embeddings from Malaviya et al. (2017) and Östling

and Tiedemann (2017) are used, which are tested for superior performance in typological feature prediction in syntax, phonology and genealogical features, respectively. Specifically, to test hypothesis (ii), we analyse the CLICS and WordNet-based language embeddings' performance on the typology feature prediction and their ability to represent the language similarity compared to typological features (cf. Section 5).

Subsequently, four datasets are created for typology feature prediction by the common set of languages, i.e., CLICS ∩ WALS, WordNet (Concept) ∩ WALS, Malaviya et al. (2017) ∩ WALS, and Östling and Tiedemann (2017) ∩ WALS. The statistics of the intersecting languages with WALS and selecting typological feature areas are shown in Table 2.

**Experimental Setup** We conduct the typology feature prediction experiments using a simple classifier consisting of a one-layer feedforward neural network with a dropout of 50%, and a softmax layer. For each feature, a multi-class classifier is trained maximally for 100 epochs. The cross-entropy loss is used to evaluate at the end of each epoch. To ensure a fair comparison, for all three datasets, as shown in Table 2, a common set of test data across the data sets is created, consisting of 74 languages. Then for each dataset, the rest of the data is split into train and dev sets. The number of data samples is very limited for each feature, as indicated in Table 2. Ten-fold cross-validation on the train-dev splits is therefore implemented to promote the performance.

To assess whether learned language embeddings capture extra semantic information, we im-

[6] The learned language embeddings are made publicly accessible in our GitHub repository https://shorturl.at/zFISY.

[7] https://doi.org/10.5281/zenodo.7385533

plement a baseline classifier with a majority vote, and a model with the same one-layer feedforward neural network structure but with an embedding layer initialized with random distribution.

## 5 Analyses and Results

**Comparing Language Embeddings** As indicated in Table 2, each feature area has an uneven distribution of features, feature labels and data samples. Hence, the macro F1 score is used to record test results for each feature, and for each feature area, the results of all the included features are averaged. For colexification-informed language embeddings, we present the results of the models with the median and best averaged macro F1-scores for each selecting feature area and the averaged results of all the feature areas.

As shown in Table 3, the baseline and the model with randomly initialized embeddings perform on par across the datasets, whereas all the colexification-informed language embeddings beat the baseline for each feature area and also on average across feature areas, and present the most performance gain in the lexicon area, i.e., 9.91 and 9.97 with WordNet Concept (best) ($W_{glove\_concat+avg}$) and CLICS (best) ($W_{prone\_concat+max}$), respectively. In contrast, the language embeddings from Malaviya et al. (2017) perform the worst for the lexicon features, while having the most performance gain in syntactic feature areas. While the language embeddings from Östling and Tiedemann (2017) perform better in lexicon feature areas compared to Malaviya et al. (2017), both best performing colexification-informed language embeddings still have two percent more performance gains. These results not only corroborate our hypothesis that the colexification-informed language embeddings capture a unique aspect, especially in semantic typological features, but also indicate that in general, leveraging semantic typology information could boost the performance of downstream tasks.

**Capturing Lexicon Typological Features** To better understand how the colexification-informed language embeddings better capture semantic typological information, we analyze the performance of lexicon feature prediction with several representative examples, as visualized in Figure 2.

The left side of Figure 2 presents the performance of CLICS (best) model and the corresponding Random model in predicting each feature (e.g.,

Number of Basic Colour Categories), the colour of the circle represents the feature values (e.g., 6-6.5 and 11), and the size of the circles indicates its proportion of the data samples for the regarding values in the train data (e.g., there are more data samples for the feature value "11" than "6-6.5"). Overall, CLICS outperforms Random in almost each feature value across lexicon features. In comparison, CLICS excels at the uneven distribution of train data samples. For instance, for features "Number of Non-Derived Basic Colour Categories" and "Number of Basic Colour Categories", the feature values "4.5" and "11" have fewer samples compared to their counterparts, while Random cannot detect them, CLICS obtained 50% and 80% performance.

Similar results are shown for the Wordnet-based models and their corresponding Random model, as shown on the right side of Figure 2. For the feature "Number of Basic Colour Categories", both WordNet and WordNet Concept models achieve the perfect score compared to the Random counterpart, which is not able to identify the minority class at all. Whereas, WordNet Concept outperforms WordNet for the feature "Number of Non-Derived Basic Colour Categories", WordNet Concept has a 100% macro F1-score with WordNet and Random failing to identify the minority class.

These results demonstrate that the models trained with colexification-informed language embeddings have learned to better capture the semantic typology information compared to randomly initialized embeddings. The language embeddings could be further fine-tuned and applied to assist other NLP applications.

**Language Similarities** Having attested that the colexification-informed language embeddings capture the semantic typological aspects of languages, we investigate how well the language similarities represented by the semantic typology features and the language embeddings correlate.

To represent languages by lexicon features, we generate a vector for each language by encoding a 13-dimensional vector with the feature values padded with -1, if the feature value is absent. The cosine similarities among the vectors are calculated. Similarly, the cosine similarities are calculated for the language embeddings. The Pearson correlation coefficient and p-value [8] for testing non-correlation are calculated between the lan-

---

[8] https://shorturl.at/rDO89

| Model | Lexicon | Complex Sentences | Nominal Categories | Simple Clauses | Average (All Features) |
|---|---|---|---|---|---|
| CLICS ∩ WALS | | | | | |
| **Baseline** | 39.85 | 21.89 | 21.94 | 26.73 | 29.82 |
| **Random** | 37.88 (-1.97) | 23.16 (+1.27) | 21.06 (-0.88) | 27.14 (+0.41) | 29.97 (+0.15) |
| **CLICS (Median)** | 41.88 (+2.03) | 27.73 (+5.84) | 26.11 (+4.17) | **29.71 (+2.98)** | 30.45 (+0.63) |
| **CLICS (Best)** | 49.76 (+9.91) | **29.32 (+7.43)** | 27.33 (+5.39) | 27.91 (+1.18) | **34.96 (+5.14)** |
| WordNet ∩ WALS | | | | | |
| **Baseline** | 37.87 | 19.26 | 24.12 | 37.67 | 33.06 |
| **Random** | 38.54 (+0.67) | 19.26 | 22.42 (-1.70) | 34.99 (-2.68) | 32.95 (-0.11) |
| **WordNet (Median)** | 36.59 (-1.28) | 23.89 (+4.63) | 28.05 (+3.93) | 37.12 (-0.56) | 34.17 (+1.11) |
| **WordNet Concept (Median)** | 39.09 (+1.23) | 25.43 (+6.17) | 27.73 (+3.61) | 37.63 (-0.04) | 34.94 (+1.88) |
| **WordNet (Best)** | 47.07 (+9.20) | 26.17 (+6.91) | 32.56 (+8.44) | **40.23 (+2.56)** | 37.11 (+4.05) |
| **WordNet Concept (Best)** | 47.84 (+9.97) | **26.52 (+7.26)** | 34.53 (+10.11) | 38.96 (+1.29) | **39.91 (+6.85)** |
| Malaviya et al. (2017) ∩ WALS | | | | | |
| **Baseline** | 34.83 | 18.94 | 21.98 | 32.76 | 31.00 |
| **Random** | 34.83 | 19.68 (+0.74) | 21.21 (-0.77) | 33.69 (+0.93) | 30.94 (-0.06) |
| **MTCELL** | 34.43 (-0.4) | 0.2549 (+6.55) | **34.74 (+12.76)** | 42.79 (10.03) | 35.14 (+4.15) |
| **MTVEC** | 21.85 (-12.98) | 23.55 (+4.61) | 25.03 (+3.05) | **36.21 (+3.45)** | 34.49 (+3.49) |
| **MTBOTH** | 31.29 (-3.54) | **29.83 (+10.89)** | 31.66 (+9.68) | 39.37 (+6.61) | **38.21 (+7.22)** |
| Östling and Tiedemann (2017) ∩ WALS | | | | | |
| **Baseline** | 35.01 | 18.99 | 20.39 | 34.44 | 31.07 |
| **Random** | 35.01 | 18.99 | 21.62(+1.23) | 34.57 (+0.13) | 30.88 (-0.19) |
| **L1** | 35.17 (+0.16) | **26.94 (+7.95)** | 25.32 (+4.93) | **37.78 (+3.34)** | 31.26 (+0.20) |
| **L2** | **42.64 (+7.63)** | 17.14 (-1.85) | **26.78 (+6.39)** | 36.02 (+1.58) | 31.80 (+0.73) |
| **L3** | 34.68 (-0.33) | 22.90 (+3.91) | 23.99 (+3.60) | 35.59 (+1.15) | **33.51 (+2.45)** |

Table 3: Test Results of Typological Feature Prediction. Results are in macro-f1 scores, numbers in brackets are the performance gains compared to the corresponding baseline, bold numbers indicate the highest performance gain compared to the corresponding baseline model, and the underlined results indicate the model with the highest performance gain per feature.

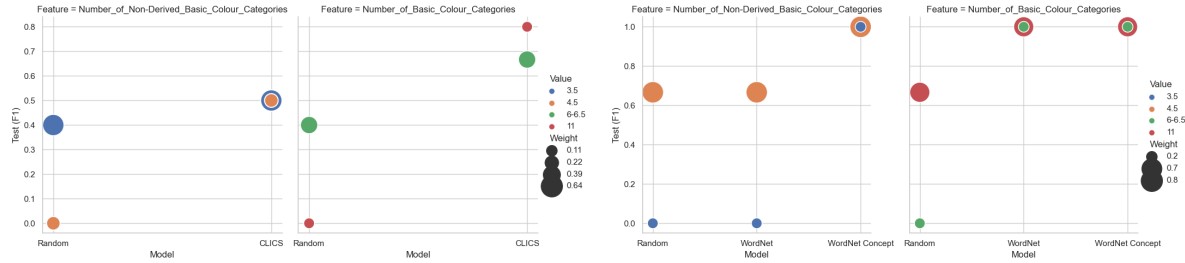

Figure 2: Performance of Predicting Lexicon Typological Features. The test results are in macro F1-scores, the colour of the circle represents the feature values, and the size of the circles indicates the size of the data samples for the regarding values in the train data.

| Language Embeddings | #Language (Pair) | Correlation Coefficient (P-value) | #Language (Pair) | Correlation Coefficient (P-value) |
|---|---|---|---|---|
| CLICS | 343 (58653) | - 0.049 (4.436e-33*) | 8 (28) | - 0.0876 (0.6575) |
| WordNet | 216 (23220) | **0.1469** (3.525e-112*) | 8 (28) | 0.7679 (1.838e-06*) |
| WordNet Concept | 216 (23220) | 0.1274 (1.339e-84*) | 8 (28) | **0.8515** (9.210e-09*) |

Table 4: Correlation between Language Similarities represented by Lexicon Typological Features and Colexification-informed Language Embeddings. * indicates that the correlation is statically significant, the numbers in bold indicate the highest correlation coefficients.

guage similarities represented by lexicon typology features and language embeddings. We present the results for the three best-performing language embeddings with both whole language sets intersected with WALS and a case study on a set of Nordic and Baltic languages, as shown in Table 4.

When tested with large sets of language pairs, i.e., 58,653 and 23,220 in CLICS and WordNet-

based, respectively, all three correlations are statistically significant, and WordNet-based language embeddings present stronger positive correlations with lexicon typological features in representing language similarities. This verifies our hypothesis that the language embeddings learned on large-scale WordNet datasets present stronger semantic typological signals than the one trained on CLICS.

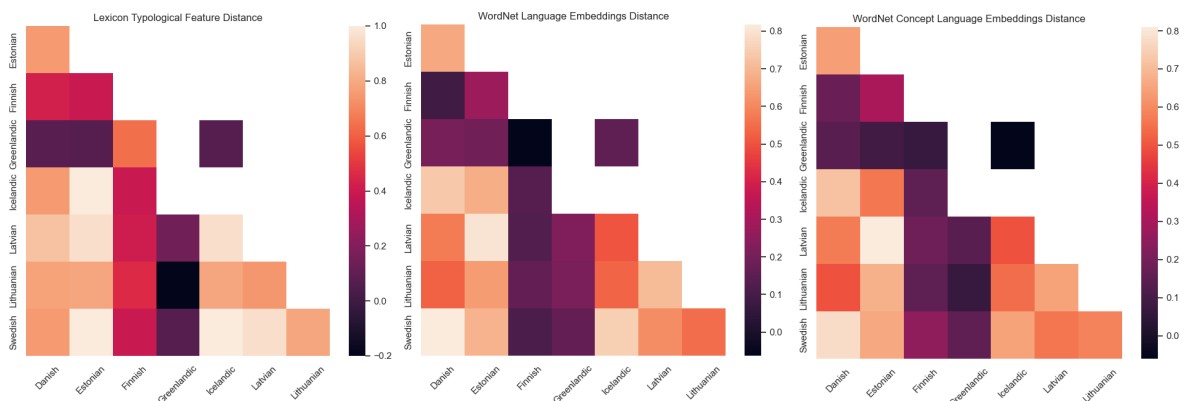

Figure 3: Language similarities represented by Lexicon Typological Features and Colexification-informed Embeddings.

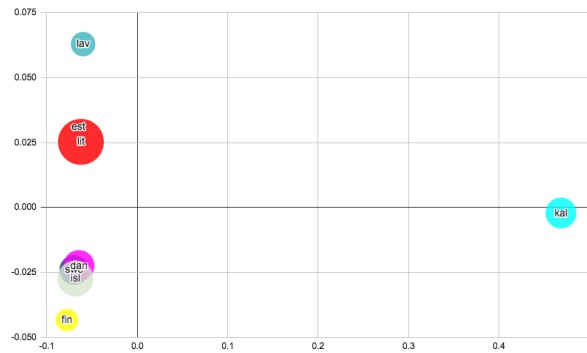

Figure 4: Language similarities represented by applying PCA on WordNet language embeddings.

The information density of the language embeddings increases with the number of incorporated synsets and colexification patterns.

A set of Nordic and Baltic languages are selected to compare further the represented language similarities. Both WordNet-based language embeddings present strong positive correlations, i.e., 0.7679 and 0.8515, respectively, and the correlations are statistically significant, as shown on the right side of Table 4. To further analyse the results, the heatmap is used to visualize the language similarities represented by lexicon features and WordNet-based language embeddings, as shown in Figure 3. The most distinctive difference is that Finnish is highly similar in terms of lexicon features compared to other languages but relatively dissimilar in terms of WordNet-based language embeddings. In this context, the WordNet-based embeddings arguably present a more realistic image of language similarities semantically.

We differentiated the WordNet Concept from

WordNet dataset, assuming that a dataset with only concepts would avoid data noises and render language embeddings able to better capture the semantic associations between languages. However, the analysed results do not corroborate the assumption. On the contrary, the language embeddings learned on all the WordNet synsets present a stronger correlation (+0.02) with lexicon typological features.

To further investigate language similarities, we apply PCA to the WordNet-based language embeddings (Figure 4). We can observe that, e.g., Scandinavian languages are clustered together, as expected. Another observation is that Finnish is relatively close to this cluster, owing to a relatively high amount of overlapping colexification patterns from language contact with Swedish, as compared to Estonian which is placed closer to one of its contact languages, Lithuanian.

## 6 Conclusion and Future Work

In this study, we have proposed a framework `Colex2Lang` to leverage colexifications to learn language representations and explored the potential of using semantic typology in NLP. A large-scale synset graph is constructed using WordNet source from Babelnet, and three datasets of colexification are processed including CLICS. Subsequently, within each dataset, 24 language embeddings variants are learned, and further evaluated and analysed by typology feature prediction and modelling language similarity. We have demonstrated, at a large scale, that colexification-informed language embeddings capture a distinctive aspect of languages in terms of semantic typology, and the data scope of the curated synsets

affects the performance of applying language embeddings. Furthermore, the analysis of representing language similarities by using learned language embeddings illustrates a realistic approach.

A large body of research has demonstrated the use of syntactic, genealogical and geographical information from linguistic typology to learn language representations, model language similarities, and further improve transfer learning performance in downstream tasks in NLP. Our work is the first attempt to learn language representations and model language similarity by leveraging semantic typology. The framework provides a strong benchmark for further research in this direction.

For future work, the benefits of applying colexification-informed language embeddings will be extensively explored. Multilingual semantic parsing is a clear candidate, where a cross-lingual signal based on colexifications may prove useful. The language similarities represented by colexifications could further inspire multilingual transfer learning, as in leveraging high-resource languages with dedicated lexical data to improve performance in semantically similar low-resource languages.

## Acknowledgments

This work is supported by the Carlsberg Foundation under a *Semper Ardens: Accelerate* career grant held by JB, entitled 'Multilingual Modelling for Resource-Poor Languages', grant code *CF21-0454*. We are furthermore grateful to Heather Lent for her insightful comments on earlier versions of this manuscript, and to Esther Ploeger for assistance in extracting typological feature prediction data from WALS.

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

# A Construction of Colexification Graph

We adopt the algorithm presented in Harvill et al. (2022) to construct a large-scale synset graph from WordNet synsets for our study (cf. Section 3). The difference in our approach lies in the addition of $G_s$ at line 3 and line 9, as shown in Algorithm 1. $G_s$ affords the constructions of colexification embeddings and language embeddings after obtaining synset embeddings trained on $G$ with node embeddings algorithms (cf. Section 3).

---

**Algorithm 1** Construction of Colexification Graph: Given a set of languages L and corresponding vocabularies V, create graph edges between all colexified synset pairs (nodes), consisting of the set of tuples of lemmas and their language.

---

1: **function** CONSTRUCTGRAPH($L$,$V$)
2: $\quad CSP \leftarrow \{\}$ $\qquad \triangleright$ Colexified Synset Pairs
3: $\quad G_s \leftarrow \textbf{graph}$
4: $\quad \textbf{for } l \in L \textbf{ do}$
5: $\quad\quad \textbf{for } x \in V_l \textbf{ do}$
6: $\quad\quad\quad \textbf{if } |S_x| \geq 2 \textbf{ then}$
7: $\quad\quad\quad\quad \textbf{for } \{s_1, s_2\} \in \binom{S_x}{2} \textbf{ do}$
8: $\quad\quad\quad\quad\quad CSP \leftarrow CSP \cup \{s_i, s_j\}$
9: $\quad\quad\quad\quad\quad G_s(s_1, s_2) \leftarrow \{x, l\}$
10: $\quad\quad\quad\quad \textbf{end for}$
11: $\quad\quad\quad \textbf{end if}$
12: $\quad\quad \textbf{end for}$
13: $\quad \textbf{end for}$
14: $\quad G \leftarrow \textbf{graph}$
15: $\quad \textbf{for } s_1, s_2 \in CSP \textbf{ do}$
16: $\quad\quad G(s_1, s_2) \leftarrow 1$
17: $\quad \textbf{end for}$
18: $\quad \textbf{return } G$
19: $\quad \textbf{return } G_s$
20: **end function**

---