# OpenReview forum: "Colex2Lang: Language Embeddings from Semantic Typology"
_NoDaLiDa/2023/Conference — NoDaLiDa 2023_

### Official Review · Reviewer_MwvX · 2023-03-09
**An interesting paper but Github code is missing**

**Rating:** 7
**Confidence:** 4

**Review:**

Codex2Lang: Language Embedding from Semantic Typology is an interesting paper; unfortunately, I could not see the associated GitHub repository for anonymity reasons. Therefore the way the authors experiment is not entirely clear. Here are some things that could be improved:
On page 1, the claim that Wordnet is a child of big data and has been developed in recent years is not true.

Given the importance of the synset graph algorithm in the paper, the procedure for constructing it should be explicitly given. Instead, the authors provide a reference to an article by Harvill.

On page 4, the GloVe algorithm is mentioned, but the authors meant the modified variant of the GloVe that works on graphs. This fact should be made explicit. To adapt GloVe for use with graphs, the word co-occurrence matrix used in the original GloVe algorithm would be replaced with an appropriate graph representation. For example, the adjacency matrix or an edge list can be used to represent the graph. Instead of counting word co-occurrences, the frequency of node co-occurrences in the graph would be used to compute the co-occurrence matrix.

The paper's main result is that the colexification model language embeddings improve the performance of predicting some lexico-semantic typological features. But the authors should venture to say why this is the case. What features of the colexification help improve the prediction performance? What are they capturing about the multilingual language semantics, and how is this related to semantic typological features?

The authors repeatedly state that the colexification model language embeddings can be further tuned and applied for downstream tasks and that these embeddings will improve the accuracy of these applications. In the Conclusion section, you mention the semantic multilingual semantic parsing as a possible application. However, you need to give evidence to support this claim. It would be more appropriate to state this as a hypothesis that can be proved or disproved in future work.

**Paper Type:**

Long paper

---

### Official Review · Reviewer_U9cd · 2023-03-10
**A motivated and novel study on the semantic typology of languages**

**Rating:** 7
**Confidence:** 3

**Review:**

The study introduces a novel approach to the semantic typology of languages based on cross-lingual colexification patterns (i.e., a word form refers to different concepts in a language). This approach is then used for training semantically informed language embeddings which are tested in the typological feature prediction and language similarity benchmarks. Experimental results on a relatively large test environment show promising performance on a variety of probing tasks.

Overall, the paper is motivated and well-presented. The experimental design is accurate, and the experiment setup seems appropriate.

**Paper Type:**

Long paper

---

### Decision · Program_Chairs · 2023-03-17

Accept